Influences of environment, human activity, and climate on the invasion of Ageratina adenophora (Spreng.) in Southwest China

Zhang Xiaojuan 1 2 3
Wang Guoyan wangguoyan@cdut.edu.cn 1 3
Peng Peihao 2 3 4
Zhou Yongxiu 5
Chen Zhuo 2 3
Feng Yu 2
Wang Yanru 2
Shi Songlin 2 3
Li Jingji 1 3
1 State Key Laboratory of Geohazard Prevention and Geoenvironment Protection, Chengdu University of Technology Chengdu , Sichuan , China
2 College of Earth Sciences, Chengdu University of Technology , Chengdu , Sichuan , China
3 Institute of Ecological Resources and Landscape, Chengdu University of Technology , Chengdu , Sichuan , China
4 College of Tourism and Urban-Rural Planning, Chengdu University of Technology , Chengdu , Sichuan , China
5 College of Geophysics, Chengdu University of Technolog , Chengdu , Sichuan , China
Dąbrowski Piotr
Electronic publication date: 2023 Mar 9
Publication date: 2023
Volume: 11
Electronic Location ID: e14902
Received 2022 Aug 4; Accepted 2023 Jan 24
Copyright: ©2023 Zhang et al.
Copyright year: 2023
Copyright holder: Zhang et al.
License: This is an open access article distributed under the terms of the Creative Commons Attribution License, which permits unrestricted use, distribution, reproduction and adaptation in any medium and for any purpose provided that it is properly attributed. For attribution, the original author(s), title, publication source (PeerJ) and either DOI or URL of the article must be cited.
License URL: https://creativecommons.org/licenses/by/4.0/

Keywords: Invasive plant, Ageratina adenophora, Multiple factors, Community structure, Forest management

Funding: National Natural Science Foundation of China 31860123 31560153 Strategic Priority Research Program of the Chinese Academy of Sciences XDA26010101 Second Tibetan Plateau Scientific Expedition and Research (STEP) program 2019QZKK0301 This work was supported by the National Natural Science Foundation of China (31860123, 31560153), Strategic Priority Research Program of the Chinese Academy of Sciences (XDA26010101), the Second Tibetan Plateau Scientific Expedition and Research (STEP) program (2019QZKK0301). The funders had no role in study design, data collection and analysis, decision to publish, or preparation of the manuscript.

==============================
With economic and social globalization, invasive alien species have significantly threatened local ecological security. Identifying the invasive mechanisms of invasive alien species can aid in preventing species invasions and protecting local ecological and economic security. As a globally invasive plant, Ageratina adenophora (Asteraceae) has spread to many parts of the world and had a seriously impacted the ecology and economy of its invaded areas. Using observational data and Landsat OLI images in an arid valley region in southwest China, this study examined how climate, human activity and environmental factors influence the invasion of A. adenophora and its underlying mechanism. Our results showed that the invasion abundance of A. adenophora was significantly affected by environmental factors (the relative importance was 87.2%), but was less influenced by human activity and climate factors (the relative importance was 2% and 10.8%, respectively). The A. adenophora abundance significantly decreased with aspect, community canopy density, shrub layer coverage, herb layer coverage, Simpson diversity index of shrub and herb layers, the shortest distance to residential areas and temperature seasonality, whereas it increased with soil moisture, temperature annual range, precipitation of wettest month and precipitation of driest month. We conclude that biotic competition is the most influential factor in the invasion of this plant in the arid valley regions. Our results are of great significance for invasion prevention and forest conservation and management in southwest China. Our work emphasized that optimizing the community structure, such as by increasing canopy and shrub coverage and species biodiversity, may help control and mitigate the A. adenophora invasion in southwest China.

Introduction

With the increasing globalization of the economy, biological invasions have become a serious global problem (Alpert, Bone & Holzapfel, 2000; De Camargo, Cunico & Gomes, 2022; Hulme, 2010; Pathak et al., 2019). There is a mounting consensus that invasive alien species constitute one of the most serious and rapidly growing threats to biodiversity, ecosystem function and services, resource availability, economic sustainability, etc (Levine et al., 2003; Livingstone, Isaac & Cadotte, 2020; Pyšek et al., 2004; Valentine Lafond et al., 2020). Invasive alien species often exhibit newly evolved traits, such as increased competitive and dispersal abilities in new habitats. Many countries regard invasive alien species as a more serious threat than climate change. Such invasive species have caused the extinction of native plants and animals, degradation of rare and threatened ecosystems and ecological communities, and so on (Díaz et al., 2015; IPBES, 2018). Studying the multiple factors where the alien plants have invaded helps in understanding the potential invasive drivers, and prediction of the sites most vulnerable to invasive plants, thus providing a basis for preventing and controlling plant invasion (Inés et al., 2009; Ward et al., 2020b).

There is growing interest in unraveling the environmental factors affecting the distribution and abundance of the plants involved (Catford, Jansson & Nilsson, 2010; Catford et al., 2011; Davis, Grime & Thompson, 2000; González-Moreno et al., 2013). Although significant advances have been made, the effects of multiple factors on plant invasions of forest ecosystems are still poorly understood. These factors may influence processes ranging from local to regional, such as resource availability and biological interactions (Levine, 2000; Pyšek et al., 2010). On a regional scale, the climate is the primary limiting factor for invasion, and the effects of climate have been the most frequently studied at a large scale, as climate patterns determine macro-environmental conditions for the distribution of species from the continental to regional scales (Milbau et al., 2009; Pearson & Dawson, 2003). The effects of anthropogenic disturbances are typically reflected at the propagule dissemination stage (González-Moreno et al., 2014), for example, roads and rivers facilitate the spread of alien plants by providing corridors for invasion (Forman & Alexander, 1998; Ward et al., 2020a). Regions with high human influence have an increased likelihood of non-native plant arrival and establishment (Kueffer et al., 2010; Pyšek et al., 2020). Environmental characteristics mainly determine what is grown locally (Ibanez et al., 2019). Terrain factors alter precipitation and solar radiation redistribution, resulting in different microclimate patterns to accelerate plant invasion (Wang et al., 2018). Canopy gaps favor alien weeds, which can more readily respond to light condition changes under the forest canopy (Qi et al., 2014; West, Matlaga & Davis, 2014).

Crofton weed, Ageratina adenophora (Spreng.) R.M.King & H. Rob is a worldwide invasive plant (Dong et al., 2008a; Poudel et al., 2019). As a perennial semi-shrubby herb of the genus Ageratina (Asteraceae), it is native to Central America from Mexico to Costa Rica (Auld, 1969), and was introduced to Europe, Australia and Asia as a garden plant in the 1950s and 1960s, and then it has become widespread in many tropical and subtropical regions of the world due to its high environmental tolerance and wide ecological adaptability (Poudel et al., 2019). A. adenophora invaded southern of China from Myanmar along the Dian–Myanmar International Road through border communication and traffic transportation around 1940s (Sun, Lu & Sang, 2004; Wang & Wang, 2006). It has now infiltrated and threatened native ecosystems in more than a third of China, ranking first on the list of alien species in the country (Wan et al., 2010; Wang et al., 2017), and is spreading east and north at a rate of about 30–50 km per year (Qiang, 1998; Wang et al., 2011). It has been reported that A. adenophora has caused serious losses to local agricultural, forestry, and livestock production, and it poses a serious threat to local species diversity and ecological security as a result of its competition with native plants to crowd out them and form single-advantaged communities (Nie et al., 2012; Sun, Lu & Sang, 2004). Most previous studies have focused on aspects such as geographical distribution, biological characteristics, allelopathy, soil microorganisms and hazards, but there are few reports on the mechanisms by which A. adenophora affects forest ecosystems from multiple factors of ecological and environmental conditions. Niu et al. (2007) studied the effect of A. adenophora invasion on soil communities under forests, and their findings indicate that compared with the resident natives, A. adenophora is more positively affected by the soil community associated with native communities, and once the invader becomes established, it further alters the soil community in a way that favors itself and inhibits natives, thereby promoting the invasion. Lu & Ma (2006) indicated that roads and streams were the main conduits for the spread of Crofton weed in southwest China, and reduced native species cover could facilitate invasion by Crofton weed in habitats along roads and streams. Compared with previous studies, our research aims to examine multiple factors of invasion of forest communities by the invasive alien plant, such as from the local forest community structure where the alien invasive plant A. adenophora is located, to the regional human activity interference, and to the macroclimate conditions. Our findings can effectively prevent and control alien invasive plants, and provide the scientific basis for managing forest resources and conserving biodiversity, thus filling the knowledge gap on invasive alien plants invading invasive alien plants in southwest China and similar ecosystem types.

The invasion of forest communities in recent years has caused serious damage to the local natural landscape, ecosystem function and social economy, so it is urgent to explore the mechanisms affecting the invasion of A. adenophora. The purpose of this study was to examine the potential mechanisms of the invasion of A. adenophora in the arid valley area of southwestern China, and answer the following questions: (i) How do the environment, human activity, and climate factors influence the invasion of forest communities by invasive alien plants, and (ii) which factors contribute the most to explaining this plant’s invasiveness? Answers to these questions will help to develop action plans and better inform management strategies to prevent the invasive alien plants in southwest China and similar ecosystem types.

Materials and Methods

Study area

The arid valley region of southwestern China is one of the most severely invaded areas by A. adenophora, which has steep climatic gradients from valleys to high mountains, and its climate zone ranges from south subtropical to temperate. The study area was located in 26°03′N ∼29°27′N, 100°15′E ∼103°53′E, with an altitude of 305–5,958 m. Annual mean temperatures range between 14 and 21 °C, and annual precipitation ranges between 760 and 1,200 mm. The rainy season from May to October accounts for 90% of the annual rainfall, and the frost-free period exceeds 300 days in a year. The geological structure of the study area is complex, with Yungui Plateau in the East, Hengduan Mountains in the west, Jinshajiang Gorge in the South, and Sichuan Basin in the north. Its mountainous landscape is formed by four major fault zones (the Jinhe - Qinghe Fault Zone, the Panzhihua Fault Zone, the Xigeda Fault Zone, and the Anning River Fault Zone). Human activities have a profound impact on the study area. In the past, people obtained wood primarily through forest exploitation and logging, and carried out forest management by burning forest litter to promote seedling regeneration. These have resulted in the loss of native plants and the serious invasion of alien plants.

Field data

In winter 2019, we conducted a field survey from north to south in an arid valley in southwestern China (Fig. 1). We selected Pinus yunnanensis forest communities with different invasion abundances of A. adenophora for investigation. A total of 40 fixed plots were surveyed, in each 10 × 10 m2 tree plot, we surveyed two 5 × 5 m2 shrub diagonal plots, and in each shrub plot, we randomly surveyed two 1 × 1m2 herb plots (Fig. 1). Each plot was selected to be representative of local plant growth as possible. We compiled complete species lists of all vascular plant species in the plots; the plants were identified by plant taxonomists. We recorded the quantities, diameter at breast height (DBH) and height of all tree species, and the species types, quantities, height, coverage, crown size and bush diameter of the shrub layer and herb layer (excluding A.adenophora). Community canopy density was measured by a Canopy Analyzer, the topographic factors (altitude, slope and aspect) were measured using Compass Gauge and topographic relief was calculated based on DEM (30 m resolution). In each tree plot, we randomly measure the soil moisture and the soil temperature five times by using a Soil Moisture Meter (Takeme-10, Zheqin Technology Co., Ltd., Dalian, China), with a five cm depth, and in total, we collected 200 soil moisture and temperature values in 40 sample plots.

Figure 1 Location of the studied forest plots and design of the plot layout in Southwestern China.

(A) shows the topographical map of the survey area and the design of the plot layout, (B), (C), and (D) represent the three categories of abundance (low, medium, and high) of A. adenophora in the forest undergrowth respectively.

Predictor variables

The predictor variables are divided into three groups: environment, human activity, and climate variables (Table 1). The environment variable were mainly obtained from the field survey. Human activity variables were downloaded from open street map (https://www.openstreetmap.org/). Climate variable data were all obtained from the WorldClim dataset (https://www.worldclim.org/). All predictor variables are exacted by sampling points (x, y).

Table 1 Predictor variables of A. adenophora invasion abundance in forest communities.

Predictor variables	Data source	Unit	
Environment variables			
altitude	Field survey (measured by Compass Gauge)	m	
slope	∘	
aspect		
topographic relief	DEM (http://www.gscloud.cn)		
burn intensity	Field survey		
soil moisture	Field survey (measured by hygrometer)	%	
soil temperature	Field survey (measured by thermometer)	°C	
community canopy density	Field survey (measured by canopy analyzer)	%	
shrub layer coverage	Field survey (estimated by plant taxonomy experts)	%	
herb layer coverage	%	
Simpson diversity index of tree layer	Field survey ((D=1−∑pi2 where pi is the ratio of the importance value the ni of species i to the sum of the importance values of all species)		
Simpson diversity index of shrub layer		
Simpson diversity index of herb layer		
Human activity variables			
shortest distance to roads	open street map (https://www.openstreetmap.org/)	km	
shortest distance to waters	km	
shortest distance to residential	km	
Climate variables			
temperature annual range	WorldClim (https://www.worldclim.org/)	°C	
precipitation seasonality	%	
precipitation of driest month	mm	
precipitation of wettest month	mm	
temperature seasonality	°C	
annual precipitation	mm	
mean temperature of coldest quarter	°C	

Statistical analysis

To avoid model overfitting caused by multicollinearity between the variables (Dormann et al., 2013), we checked the collinearity among the selection of predictors by Spearman correlation tests (Dormann et al., 2013). We filtered and deleted independent variables with high correlations (ρ > 0.75, Appendix 1). For instance, if the absolute value of the cross-correlation coefficient between two variables exceeded 0.75, only the variable that captured the best ecological meaning and explanatory power was selected (Table 1). Taking human activity variables into account, we found that the shortest distance to waters has a strong covariance to the precipitation factors, therefore, for the full models, we used the shortest distance to roads, the shortest distance to waters and the shortest distance to residential. With regard to climate variables, we selected temperature annual range, precipitation seasonality, precipitation of driest month, precipitation of wettest month, temperature seasonality, annual precipitation and mean temperature of coldest quarter. (ρ < 0.75, Table 1, Appendix 1). Spatial autocorrelation among the plots was tested by Moran’s I, and the results indicate that the sampling plots are independent of each other (P = 0.21), which conforms to the modeling premise.

We used generalized linear models (GLMs) (Burnham & Anderson, 2004) to analyze the effects of climate, human activities, and environment variables on the A. adenophora abundance. The reduction or increase of the linear model deviation was evaluated separately through the Akaike Information Criterion (AIC) (Table 2, Fig. 2) (Turner & Crawley, 1994; Venables & Ripley, 1998). We selected the best model (smallest AIC) for each block of non-collinear predictors (i.e., environment, human activity and climate), we retained slope, aspect, soil moisture, community canopy density, shrub layer coverage, herb layer coverage, Simpson diversity index values of the shrub and herb layers, the shortest distance to roads, the shortest distance to residential, temperature seasonality, temperature annual range, precipitation of wettest month, precipitation of driest month in the final model. In addition, we evaluate the relative contribution of the predictor variables to the response variable by calculating the relative weight (R2 in Table 2), the larger the value was, the greater the contribution (Jiao et al., 2020; Nakagawa & Schielzeth, 2013).

Table 2 Influences of environment, human activity, and climate on the invasion of Ageratina adenophora.

Gaussian distribution	Estimate	Std.Error	t value	Pr(>—t—)	R2(%)	2.5%CI	97.5%CI	
(Intercept)	0.00	0.06	0.00	1	91.0	−0.12	0.12	
slope	0.11	0.08	1.48	0.1519	1.8	−0.04	0.26	
aspect	−0.15	0.07	−2.04	0.0521.	3.3	−0.29	−0.01	
soil moisture	0.33	0.08	4.07	0.0004***	8.8	0.17	0.49	
community canopy density	−0.34	0.07	−4.90	0.0000***	12.1	−0.48	−0.20	
shrub layer coverage	−0.25	0.08	−3.22	0.0035**	21.0	−0.41	−0.10	
herb layer coverage	−0.22	0.08	−2.71	0.0121*	5.9	−0.38	−0.06	
Simpson diversity index of shrub layer	−0.15	0.08	−1.96	0.0618.	4.7	−0.30	0.00	
Simpson diversity index of herb layer	−0.42	0.09	−4.90	0.0000***	29.6	−0.59	−0.25	
shortest distance to roads	0.11	0.09	1.24	0.2249	0.4	−0.06	0.28	
shortest distance to residential	−0.41	0.12	−3.33	0.0027**	1.6	−0.65	−0.17	
temperature annual range	0.86	0.20	4.37	0.0002***	3.2	0.48	1.25	
precipitation of driest month	0.25	0.08	3.10	0.0048**	0.7	0.09	0.41	
precipitation of wettest month	0.75	0.18	4.16	0.0003***	2.3	0.40	1.10	
temperature seasonality	−1.17	0.32	−3.69	0.0011**	4.6	−1.79	−0.55	
Notes.

The asterisks indicate predictor variables in the model that passed the significance test, ‘*’ indicates P < 0.05, ‘**’ indicates P < 0.01, ‘***’ indicates P < 0.001, ‘.’ indicates P < 0.1, ‘ ’ indicates P > 0.1.

Figure 2 Effects of predictor variables on the A. adenophora abundance.

The asterisks indicate predictor variables in the model that passed the significance test, ‘*’ indicates P < 0.05, ‘**’ indicates P < 0.01, ‘***’ indicates P < 0.001, ‘.’ indicates P < 0.1, ‘ ’ indicates P > 0.1.

We used the variance-partition method to examine the alone and shared effects of different groups of variables on variation in the response variable (Legendre, 2008; Mood, 1969). Assuming that the deviance is a good measure of the variation in response variables explained by predictor variables, we set up environment alone, human activity alone, climate alone variables, and the three groups of variables in combination to examine the alone and shared effects on A. adenophora abundance (Carrete et al., 2007).

All statistical analyses were performed with the R-CRAN software (R Development Core Team, 2019). We tested the correlation of predictor variables using the PerformanceAnalytics package (Peterson et al., 2014), the package AER for generalized linear models (Kleiber & Zeileis, 2008) and the package VEGAN as the base code for deviance-partition (Oksanen et al., 2013).

Results

Across all the plots, we found 110 plant species belonging to 44 families and 97 genera (Appendix 2); there were 10 families, 15 genera and 18 species of trees, mainly including P. yunnanensis, Quercus variabilis, Sapindus mukorossi, Quercus aliena, and Lyonia ovalifolia. There were 42 shrub species belonging to 24 families and 37 genera, mainly Quercus monimotricha, Vernonia esculenta, Rosa sweginzowii, Prinsepia utilis, and Rubus coreanus. There were 54 herb species in 22 families and 52 genera, mainly A. adenophora, Heteropogon contortus, Ainsliaea yunnanensis, Artemisia sacrorum, and Deyeuxia pyramidalis. Apart from the invasive alien plant A. adenophora, there are four alien plants on the plots where we surveyed in the field. We include one tree, Eucalyptus robusta, and three herbs, Bidens pilosa, Crassocephalum crepidioides, and Galinsoga parviflora. We discuss the relationship between A. adenophora and other plants in this study, but we do not focus on interactions between native and alien plants in this study due to alien plants’ low frequency and low abundance in all plots.

Effects of environment, human activity and climatic factors on the A. adenophora abundance

Among the 13 environment variables, slope, aspect, soil moisture, community canopy density, shrub layer coverage, herb layer coverage, Simpson diversity index values of the shrub and herb layers were retained in the final model. We found that the slope had no significant effect on the A. adenophora invasion abundance (P > 0.05), and the soil moisture factor showed a positive effect on this invasive alien plant abundance (P < 0.001), while the aspect, community canopy density, shrub layer coverage, herb layer coverage, Simpson diversity index of shrub layer, Simpson diversity index of herb layer and the A. adenophora invasion abundance showed a negative impact on it.

For the three human activity variables of the shortest distance to roads, the shortest distance to waters and the shortest distance to residential, the shortest distance to roads and the shortest distance to residential were retained in the final model. The shortest distance to residential showed a significant negative effect on the A. adenophora abundance (P < 0.01), while the shortest distance to roads had no significant effect (P > 0.05).

Of the 7 climate variables, temperature seasonality, temperature annual range, precipitation of wettest month and precipitation of driest month were included in the final model, the temperature annual range, precipitation of wettest month, and precipitation of driest month have a positive effect on the A. Adenophora abundance, while temperature seasonality showed a negative impact.

Relative contributions of predictor variables to influence the A. adenophora abundance

In the final model, the predictor variables explain 91% of the A. adenophora abundance (Table 2). The environment variables accounted for 87.2% of the A. adenophora abundance, while the human activity and climate predictors contributed 2% and 10.8% respectively. The best environment, human activity and climate predictors explaining the A. adenophora abundance were the Simpson’s diversity index of the herb layer, shrub layer coverage and community canopy density (R2 = 29.6%, 21%, 12.1%). Simpson’s diversity index of the herb layer was the most important variable affecting the abundance of A. adenophora and it linearly decreases A. adenophora abundance (Fig. 3). Also, shrub layer coverage and community canopy density seem to have a threshold value of around 30%, where more shrub coverage and community canopy density drastically reduces A. adenophora. All other variables are not very important.

Figure 3 Multiple factors of the invasive alien plant A. adenophora in forest communities.

Plots (a, j, m) indicate the relative importance of predictor variables explaining A. adenophora abundance, and Plots (b-i, k-l, n-q) show the relationship between environment, human activity and climate variables and the A. adenophora abundance. The abbreviated letter Sdh represents Simpson’s diversity index of the herb layer, and the Sds is Simpson’s diversity index of the shrub layer.

The variation partitioning in the A. adenophora abundance

Variation partitioning showed that climate, human activity, and environment (Fig. 4) were able to explain 82% of the variation in A. adenophora abundance. Environment alone explained significantly more of the variance of the A. adenophora abundance (59%, Fig. 4) than human activity variables alone (4%) and climate alone (8%). The shared fractions among climate and environment explained a larger proportion of variation (23%, Fig. 4), while environment and human activity explained a small proportion (4%), which showed that the variation in A. adenophora abundance was primarily explained by environment and climate variables, human activity variables had a negligible explaining proportion on A. adenophora abundance. 18% of the model residuals may be due to unmeasured potential factors.

Figure 4 Variation partitioning results.

The three figure panels show Venn diagrams representing the partition of the variation of (X1) environment variables, (X2) human activity variables, and (X3) climate variables. The reported fractions are adjusted R2 statistics (R2).

Discussion

Plant invasions could be seen as a spatial hierarchical process where ecological factors affect invasions at different scales (McDonald & Urban, 2006; Milbau et al., 2009). For instance, regional human influence and climate might control variability in non-native species richness at the landscape scale, landscape characteristics might influence from the human activity to the local scale, while patch characteristics and habitat type influence invasion at the habitat scale. Following a hierarchical approach, the individual variable fitting and the generalized linear model structure demonstrated that habitat and climate variables had the most important influence on the invasion of A. adenophora in comparison to human disturbance. The low importance of human activity factors indicates that alien species abundance here is little controlled by road and river.

Effects of environment variables on the invasive alien plant

Just as previous studies have shown that species diversity is an important indicator of the invasibility of forest communities (Wu, Zhang & Dai, 2020), we found that the community canopy density, shrub layer coverage, herb layer coverage, Simpson’s diversity index of shrub and herb layers had a significant negative effect on A. adenophora abundance, this suggests that community structure and species diversity play a protective role in forests against alien invasions. Possibly similar to the generally observed pattern that high species diversity is less likely to be invaded than sites with low species diversity (Elton, 1977). The community structure with high species diversity is more stable and more competitive in terms of resources, thus exerting a negative influence on the alien plants’ invasion. In addition, maintaining forest coverage may be effective in reducing plant invasion, the light conditions are an important abiotic factor affecting the invasion process of A. adenophora (Song et al., 2017). In a well-developed forest community, the vertical structure of the community is clear, because the community canopy density and shrub layer coverage absorb or reflect the vast majority of solar radiation, blocking the light, and affecting the growth of A. adenophora. Zhang concluded that A. adenophora had the smallest height and cover in semi-humid evergreen broad-leaved forests, however, its height and cover increased sequentially in Eucalyptus artificial forests, P. yunnanensis forests, and wild mountain grasses (Zhang, 2013). Gómez et al. (2019) showed that the coverage of A. adenophora is significantly positively related to light intensity, but negatively correlated with tree cover and species richness, and that natural plant communities with high coverage can help to inhibit the invasion of A. adenophora. Our results found that Simpson’s diversity index of tree layer did not have a significant effect on A. adenophora abundance, this may be related to the fact that our plots were taken in areas where P. yunnanensis is dominant. Our research shows that soil moisture and aspect are also important factors influencing A. adenophora invasion, with A. adenophora being more abundant on the sunny slope than on the shady slope, and more widespread in spaces with high soil moisture conditions.

Effects of human activity disturbance on the invasive alien plant

Human activity variables were of minor importance in explaining A. adenophora invasion in the present study (Fig. 3). However, plots closer to residentials had higher A. adenophora abundance (Table 2). Some research suggests that invasion by alien species often begins in human-disturbed environments, where their propagules are unintentionally introduced or where cultivated species frequently escape (Boscutti et al., 2022; Geppert et al., 2021). Previous studies have proved that roads and rivers are the main routes of transmission of invasive alien plants (Dong et al., 2008a; Lu & Ma, 2006), while our results showed the shortest distance to water and roads were not the significant factors directly affecting the invasive alien plant A. adenophora. Probably because roads and rivers are indirectly influencing the invasion by affecting biological competition. Some previous research indicated that anthropogenic disturbances may affect some biological processes, such as promoting dispersal, eliminating competitors, changing environmental characteristics of the forest canopy, etc (Dong et al., 2008b; Parendes & Jones, 2000).

Effects of climate variables on the invasive alien plant

Next to the variables related to environment variables, the climate variables explained the largest amounts of variation in alien species abundance (Fig. 4). Among the 19 climate variables, temperature seasonality, temperature annual range, precipitation of wettest month and precipitation of driest month were able to explain 10.8% of the variation in alien species abundance (Table 2). Our results show that temperature annual range, precipitation of wettest month and precipitation of driest month can increase the invasion of A. adenophora, while temperature seasonality has a negative effect on it. The importance of climate for alien species richness demonstrated here (fragments in warmer areas with low humidity tending to have more alien species) have been reported from other habitats and regions (Pys˘ek, Jaros˘ıķ & Kuçera, 2002; Walther et al., 2002). However, the effects of many climate factors on the invasion of A. adenophora were not confirmed in this study, probably due to the following reasons: firstly, the strong phenotypic plasticity and adaptability of the A. adenophora, combined with the fact that it has been invading the region for decades, gives it a wider ecological range and better environmental tolerance, allowing it to occupy a wider geographical range and habitats. Secondly, the scale has an important influence on the results of community studies. Studying on the invasive alien species in southwest China, Lu found that species diversity influenced community invasibility at small scales (25 m2), while environmental factors influenced community invasibility at large scales (400 m2) (Lu, 2005).

Conclusion

The forest ecosystems are of great importance for socio-economic development and ecosystem services in this region, however, the strong reproductive capacity and phenotypic plasticity of A. adenophora pose a serious threat to forest community structure and ecosystem function in southwest China. Our results showed that the invasion ability of the alien plant A. adenophora to the forest ecosystems in the arid valley region of southwest China is mainly controlled by biotic factors. Under warm and humid climate conditions, the invasion abundance of A. adenophora was higher under the forest communities with lower Simpson’s diversity index of herb layer, lower shrub layer coverage and lower community canopy density. Therefore, in the context of global change, our findings can be used to develop action plans to manage the invasion of forest ecosystems. Our results suggest that increasing species diversity and community cover of trees and shrubs can help in preventing and controlling invasive alien plants in southwest China and similar ecosystem types.

This study primarily examines the effect of environmental conditions, human activities, and climatic conditions on A. adenophora, and although we attempted to consider as many important potential factors affecting the invasive process as possible, there are still some factors that have not been addressed. Further research could include tracking invasive alien species over time and collecting data on invasive species’ land use type, land ownership, soil quality, nutrients, soil microorganisms, etc.

Supplemental Information

Supplemental Information 1 Raw data and code

Click here for additional data file.

Thanks to the editors and anonymous experts for their comments on this research, and thanks to the assistance of the Institute of Ecological Resources and Landscape in field surveys and data collection.

Additional Information and Declarations

Competing Interests

Author Contributions

Data Availability

The authors declare there are no competing interests.

Xiaojuan Zhang conceived and designed the experiments, performed the experiments, analyzed the data, prepared figures and/or tables, authored or reviewed drafts of the article, and approved the final draft.

Guoyan Wang conceived and designed the experiments, performed the experiments, analyzed the data, prepared figures and/or tables, authored or reviewed drafts of the article, and approved the final draft.

Peihao Peng conceived and designed the experiments, performed the experiments, analyzed the data, prepared figures and/or tables, authored or reviewed drafts of the article, and approved the final draft.

Yongxiu Zhou performed the experiments, analyzed the data, prepared figures and/or tables, and approved the final draft.

Zhuo Chen analyzed the data, prepared figures and/or tables, and approved the final draft.

Yu Feng analyzed the data, prepared figures and/or tables, and approved the final draft.

Yanru Wang performed the experiments, prepared figures and/or tables, and approved the final draft.

Songlin Shi conceived and designed the experiments, authored or reviewed drafts of the article, and approved the final draft.

Jingji Li conceived and designed the experiments, authored or reviewed drafts of the article, and approved the final draft.

The following information was supplied regarding data availability:

The raw measurements are available as a Supplementary File.

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
