# Peer review of "Influences of environment, human activity, and climate on the invasion of Ageratina adenophora (Spreng.) in Southwest China"

_PeerJ, doi:10.7717/peerj.14902_

## Round 0.1 · original submission · Major Revisions

Dear Authors,

Your work has been assessed by three independent experts. All of them agreed that your work could be published in PeerJ, but needs to be significantly improved beforehand. Please read all of the reviewers' comments and respond to them.
With best regards,

Reviewer 1 ·

Basic reporting

.

Experimental design

.

Validity of the findings

.

Additional comments

Manuscript No: #76093 (peer j)
Title: Multi-scale factors determine the invasibility of the 1 forest communities by the alien plant Ageratina adenophora
Recommendation: Major revision

The manuscript #76093 discusses the multiscale factors governing the invisibility of the alien plant Ageratina adenophora and is interesting to the readers of peer J. Authors tried to elucidate the impact of various natural and anthropogenic drivers governing the invasability of A. adenophora. Though the presentation of the manuscripts seems interesting, the manuscript needs substantial revision to improve the overall quality and takeaways.

(1) The title seems confusing and needs to be modified. The denotation of ‘alien’ and ‘native’ is always with respect to a particular region or political geography, therefore the region must be there in the title. So, authors are requested to modify the title as “Multiscale factors affecting the invasability of the alien plant Ageratina adenophora in Southwest China”. Also use the authority name/citation after the species name.

(2) The abstract needs revision and needs a balance of both descriptive and quantitative information. At present, authors failed to present the key findings in the abstract.

(3) Though authors acknowledge the fact that anthropogenic factors are also governing the rampant invasion of most of the invasive species, the present study could not find any potential impact of anthropogenic factors like disturbance/linear intrusions (e.g. roads) on the invasibility of A. adenophora. On the other hand, they found biotic competition as the major factor. However, authors failed to analyze whether anthropogenic disturbances are altering the biotic competition or not and, in that way, indirectly impacting the invasion or not.

(4) Apart from the studied environmental factors like slope, aspect, soil moisture, community canopy density, shrub layer coverage, herb layer coverage etc., authors failed to study other important factors like soil quality, nutrients, soil microorganisms etc. The overall land use and land-ownership (abandoned, private or public) are also important factors affecting the invasion.

(5) There are no details regarding the periodicity and seasonality of the study and how long have authors studied the above associations in Southwest China.

(6) While authors reported the herb and shrub coverage, it is not clear whether the herb or shrub layer is already having any other local or exotic weeds and their interaction with the studied species.

(7) Authors may refer IPBES knowledge product on Invasive species assessment as it is having novel insights on plant invasion

(8) Most importantly, authors are requested to provide policy recommendations for managing the invasion of A. adenophora in southwest China and similar agro-ecological systems.

Reviewer 2 ·

Basic reporting

The article is written in English but in many parts, it is unclear and ambiguous.

Sufficient background and context provided

Figures need revision for content and quality, they are not appropriately described

No clear hypothesis stated

Experimental design

The submission defines the research question, which is relevant and meaningful. It lacks statements on how the study contributes to filling knowledge gaps.

There are some methodological and technical issues

More information is needed in Methods

Validity of the findings

The results section, particularly the modeling is deficient and seems the authors do not fully understand how the statistical models they used function and what information they offer. Rephrasing is needed in all these sections.
The aim of the study is to determine the relative effect of climate, human activity, and habitat features on the invasion of the alien plant A. adenophora and its underlying mechanism, however, the mechanisms are not mentioned in the discussion.

Conclusions are not limited and supported by results.

Additional comments

Major comments
The aim of the study is to determine the relative effect of climate, human activity, and habitat features on the invasion of the alien plant A. adenophora and its underlying mechanism, however, the mechanisms are not mentioned in the discussion.
My main concern is with the methods used to assess climatic and human activity variables on A. adenophora abundance. Particularly, there is a scaling/resampling issue. If you resample to the higher resolution your results will imply a much greater accuracy than the data on which they are based. Consider your WorldClim dataset at 1km. By whatever method they measure the temperature (avg/max/mean), the smallest unit (pixel) that can be differentiated from its neighbors is 1km. If you resample that to 10m, either 1) you assume all 10,000 of the resulting pixels are that same temperature when in truth maybe only one - the center, or the top left - (or none!) is, or 2) you interpolate between pixels, creating derived values not present before. Thus in both cases, you introduce false accuracy because your new subsamples were not actually measured. Any time you resample to a higher resolution, you are introducing false accuracy.
The results section, particularly the modeling is deficient and seems the authors do not fully understand how the statistical models they used function and what information they offer. Rephrasing is needed in all these sections.
I strongly recommend the authors check for English usage since several sections of the manuscript are unclear.
Minor comments
Abstract
Line 18. Please add here the Family of the species
Lines 22-28. These sentences are hard to follow! Please divide

Introduction
Line 85. I think there might be a grammatical error here “native communities than are resident natives”
Line 88-89. I am confused with the common name “Crofton weed” does it refers to Eupatorium or Ageratina here? Are both the same species that changed the taxonomical names as a whole or just some of the species? If the first case, please mention at the very beginning of the paragraph when you present the species its common name.
Lines 90-92. Here you can add brief reasoning for your statement. Why your study comprehensively analyzed the impact of invasive alien plants on forest ecosystems, such as climate, human activities, and habitat environment? Also, some rewording might be needed as you analyzed the effects of climate, human activities, and habitat environment on invasive alien plants

Materials and Methods
Lines 116-117. What do you mean by forest development? Deforestation due to land-use change? Please give more details on which forest resources were developed and which were logged.
Line 124-126. 40 fixed plots? Also, not sure if plots of herbs are nested on shrubs and if these are nested in trees plot… A diagram with the plot's design will be very useful.
Line 135. For clarity, please give the total number of points where you sampled the soil temperature and moisture
Lines 143-145. You mentioned before that the shortest distance was 0.074 km and the longest distance was 124 319 km between plots. However, your climate variable data were all obtained from the WorldClim dataset at 1 km2 resolution, and you mentioned all datasets were resampled to a resolution of 10 m. I am concerned about all these differences in scale. Particularly, for the resampling, it is fine to go from a finer to a coarser resolution, but going from a coarser to a finer resolution is full of uncertainties. How did you do this resampling? Also, the authors need to test for spatial autocorrelation among the plots, as the models used assume the samples are independent of each other. Moreover, looking at Fig 1, it looks like a watershed, hence, the areas in the uplands most have an effect on the areas in the lowlands.
Fig 1 image resolution and quality need to be improved. Also, there are a lot of names of cities that are not needed. Removing these will clean the map. What do the triangles refer to? It seems they are the centroids of the localities, which again, are not needed. The legend has a white spot in Sample. The description of the figure also needs to be improved to avoid grammatical errors, including where and when was the sampling, etc.

Results
Line 182. Are you referring to all the sampling plots?
Lines 199-200. Not sure what you mean here: “We found that the slope was not significant with the
model (P>0.05)” Please reword.
Line 217. What means “to evaluate the fit advantage of the final model”? Also, “the predictive variables explain 91% of the model”? Some rewording is needed. Variables do not explain a model, they constitute part of a model that explains the abundance of the invasive.
There is no need on using such precise percentages (0.123), one value is enough (0.1)
Lines 224-225. I do not understand this title. Please rephrase.
Line 232. The habitat factors explained the strongest independent explanation… this does not make sense. For clarity, please reword.
Line 233. “human activity variables had very little effect and climate factors explained 8% of the variation.” If this is correct as it is stated, here you have again a problem with scaling, since human activity layers have 30 m resolution and climate 1 km resolution. Many of your sampling plots probably have the same values for climatic variables.

Fig2. The figure description is very poor and it is showing the positive or negative effects, which can be significant or not.
Fig 3. What is sdh? Sds? Also why there are negative values for A. adenophora? Isn´t this the abundance?
From this figure, the interpretation is that Sdh was the most important variable affecting the abundance of A. adenophora. Sdh linearly decreases A. adenophora abundance. Also, shrub layer coverage seems to have a threshold value of around 30% where more shrub coverage drastically reduces A. adenophora. All other variables are not very important.
Discussion
Line 254. Not clear what “this difference” refers to.
Line 261. “…pattern that” instead of “of”
Line 264. What does “lighting” refers to? Electrical phenomena or light conditions.
Line 266. Avoid using obvious, please reword.
Line 270. What increases? Eucalyptus with capital E
Lines 273-277, 289-292. The sentence has been double cited (at the beginning and the end). Please remove one.
Line 279. What is adequate soil moisture? Please avoid using ambiguous terms.
Lines 285 and 443. There is an error with this citation.
Line 292. “Near”?
Line 292-293. It makes no sense to test this variable if all your plots are far away from rivers.
Line 302. You did not test for invasive inhibition. Avoid this type of statement.
Line 331. You are using GLMs, which are regressions, not correlations.
Line 334. But you assessed soil temperature and humidity. Also in your recommendations avoid throwing variables to assess without thinking about the implications of those variables. Habitat fragmentation… what would you consider habitat (i.e. which vegetation types)? Habitat for whom? For the invasive? For the native flora?

Reviewer 3 ·

Basic reporting

The manuscript submitted for review is constructed in accordance with the standards of the journal. It contains results from original field studies. Although the raw data has been provided, they should be further explained (measured parameter described/explained together with the unit in which it is described, e.g. m, m2). Please consider adding a list of trees/shrubs/herbs, (species lists of all vascular plant species in the plots), etc. in the raw data.
The manuscript seems to be written in correct and clear language. However, I suggest that you turn to a specialist who is fluent in English with a request to proofread the language and improve the text.

Although you use appropriate terminology referring to the analyzed problem, the interchangeable use of the terms is noticeable: invasive alien species, alien plants, exotic species, foreign species, weed. Please consider referring to the terminology proposed by Richardson et al. 2000 or Pyšek et al. 2004 (Taxon).

The description of the subject of the research should include the correct and accepted name of the species, e.g. according to The Plant List: Ageratina adenophora (Spreng.) R.M. King & H. Rob.
http://www.theplantlist.org/tpl/record/gcc-122109

The way of citing sources in the "References" section does not always meet the journal's standards. Not all quotes are complete. Please verify and complete this part of the manuscript.

Experimental design

Please explain my doubts regarding the relationship between the research questions and the methodology used. Doubt especially concerns the question: (i) to what spatial extent do habitat, human activity, and regional characteristics have maximum influence on the plant’s abundance? The question relates to a wide range of conditions in which the studied species occurs, while the research concerned the occurrence of the species in forest communities.

I have one general doubt related to the way of obtaining field data for analysis.
The description of the field studies and the collected surface data is not precise and clear. In the text, we read that data from 40 samples (plots ?) were collected, and 21 (?) are marked on the map. Were the areas located in one type of forest community or different? Without detailed data, it is difficult to assess the results obtained.

Descriptions of methods were included in the "Results" section - in my opinion, they should be moved to the "Materials and methods" section to increase the clarity of the text.

Additionally, please consider adding an appropriate map to the "Study area" part of the text to illustrate the location of the study area. Fig. 1 does not indicate exactly the boundaries of the research area.

Validity of the findings

Unfortunately, due to the imprecise description of the methods used (also the lack of some raw data), it is difficult to assess the chapter "Results" and the discussion.

Additional comments

I would like to point out other minor inaccuracies that I found in the text:

Line 67 - you indicate that the described species is „a worldwide invasive plant”, without quoting the sources.
Line 76-77: Information that „A. adenophora was transmitted to China through natural dispersal from the China-Myanmar border in the 1940s” - should be clarified - does it mean that it was introduced into a specific territory of China and spread from there spontaneously without human intervention. Do you know by what pathways?
Line 105-107: Please correct the sentence (it is long) in terms of style.
Source Citing Notes:
Line 53 - you quote: Levine & J. 2000; Pysek et al. 2020 - in the first case, the name of the second author is missing; in the second case it should be: Pyšek et al.
Line 82-83: „Hong-bang Niu et al. studied…” - there seems to be an incorrectly cited source; shouldn't there be: Niu et al. (2007) studied……
Line 87: the same as above: „Zhijun Lu et al. indicated…” – Lu and Ma (2006) indicated…
Line 90-92: How should the sentence „Compare with earlier single-scale studies, our study comprehensively analyzed the impact of invasive alien plants on forest ecosystems, such as climate, human activities, and habitat environment” be interpreted? Did you mean IAS in general or one species tested – Ageratina adenophora?
Line 285: SK et al. 2008 ??? - initials were given instead of the surname

---

## Round 0.2 · Minor Revisions

Dear Authors,

One of the reviewers has contributed comments on the revised version of your work, I kindly ask you to read the reviewer's comments and respond to them.
With best regards,

Reviewer 3 ·

Basic reporting

Dear Authors,

I have reviewed the revised version of the manuscript and supplementary materials.
In my opinion, the manuscript was significantly revised and supplemented (including additional raw data) as suggested in the review.
After the linguistic correction, the readability and quality of the text increased.
Changes in nomenclature and terminology that I have suggested have been applied. In a few cases (lines 55, 69, and 296) the term "exotic" species remained. I understand that this happened when citing other authors.

Experimental design

The description of the research area and the methodology used have been verified and supplemented. At the moment, the description has become more understandable to the reader.
Figure 1 has been corrected and supplemented - thus in this form, it better illustrates the description.
I have only a slight doubt about its description: "(d) represent the three invasive abundances (low, medium, and high) of A. adenophora under the forests respectively" - shouldn't it be:

represent the three categories of abundance (low, medium, and high) of A. adenophora in the forest undergrowth respectively (?)

Validity of the findings

The introduced corrections and additions in this chapter and in the methodological part significantly influenced the quality of this chapter. I think this article – after making corrections - meets the standards.
Please consider supplementing the results and/or adding a comment explaining whether there were other alien/invasive species on the plots where the data/observations were collected, apart from the analyzed IAS. if so, you may consider including this information in the added list of species.

Additional comments

Thank you for making the suggested minor corrections and additions.

---

## Round 0.3 · accepted · Accept

Dear Dr. Zhang

I am of the opinion that your work in its current form can be published in PeerJ. My congratulations!

With best regards,